Effects of invasion history on physiological responses to immune system activation in invasive Australian cane toads

Selechnik Daniel 1 danselechnik@gmail.com
West Andrea J. 2
Brown Gregory P. 1
Fanson Kerry V. 2
Addison BriAnne 2
Rollins Lee A. 2
Shine Richard 1
1 School of Life and Environmental Sciences (SOLES), University of Sydney , Sydney, NSW , Australia
2 Centre for Integrative Ecology, School of Life & Environmental Sciences (LES), Deakin University , Geelong, VIC , Australia
Measey John
Electronic publication date: 2017 Oct 6
Publication date: 2017
Volume: 5
Electronic Location ID: e3856
Received 2017 Aug 4; Accepted 2017 Sep 6
Copyright: © 2017 Selechnik et al.
Copyright year: 2017
Copyright holder: Selechnik et al.
License: This is an open access article distributed under the terms of the Creative Commons Attribution License, which permits unrestricted use, distribution, reproduction and adaptation in any medium and for any purpose provided that it is properly attributed. For attribution, the original author(s), title, publication source (PeerJ) and either DOI or URL of the article must be cited.
License URL: https://creativecommons.org/licenses/by/4.0/

Keywords: Rhinella marina, Eco-immunology, Phagocytosis, Cane toad, Invasive species

Funding: Australian Research Council FL12010:0074 and DE150101393 Equity Trustees Charitable Foundation [Holsworth Wildlife Research Endowment] This work was supported by the Australian Research Council [FL12010:0074, DE150101393] and the Equity Trustees Charitable Foundation [Holsworth Wildlife Research Endowment]. The funders had no role in study design, data collection and analysis, decision to publish, or preparation of the manuscript.

==============================
The cane toad (Rhinella marina) has undergone rapid evolution during its invasion of tropical Australia. Toads from invasion front populations (in Western Australia) have been reported to exhibit a stronger baseline phagocytic immune response than do conspecifics from range core populations (in Queensland). To explore this difference, we injected wild-caught toads from both areas with the experimental antigen lipopolysaccharide (LPS, to mimic bacterial infection) and measured whole-blood phagocytosis. Because the hypothalamic-pituitary-adrenal axis is stimulated by infection (and may influence immune responses), we measured glucocorticoid response through urinary corticosterone levels. Relative to injection of a control (phosphate-buffered saline), LPS injection increased both phagocytosis and the proportion of neutrophils in the blood. However, responses were similar in toads from both populations. This null result may reflect the ubiquity of bacterial risks across the toad’s invaded range; utilization of this immune pathway may not have altered during the process of invasion. LPS injection also induced a reduction in urinary corticosterone levels, perhaps as a result of chronic stress.

Introduction

Eco-immunological theory predicts that successful invaders will display reduced investment in components of the immune system that produce excessive inflammation, and/or are energetically expensive (Lee & Klasing, 2004; White, Perkins & Dunn, 2012). This prediction is based on the enemy release hypothesis (Colautti et al., 2004): the supposition that invasive hosts lose many co-evolved enemies after translocation (Allendorf, 2003; Torchin, Lafferty & Kuris, 2001), potentially reducing pathogen-mediated selection pressures (Lee & Klasing, 2004; White, Perkins & Dunn, 2012). Also, the energetic costs of mounting a strong immune response may reduce the host’s ability to survive, grow, reproduce, and disperse (Hart, 1988; Klein & Nelson, 1999; Llewelyn et al., 2010); these factors influence invasion success (Chapple, Simmonds & Wong, 2012; Cote et al., 2010). However, such a down-regulation of immune function may render invaders susceptible to infection by novel pathogens and parasites in their introduced range (Hamrick, Godt & Sherman-Broyles, 1992); for this reason, invaders are also predicted to elevate investment into less energetically costly components of the immune system (Lee & Klasing, 2004).

Components of anti-microbial activity within the innate immune system differ in the amount of energy that they require and inflammation that they cause. Systemic mechanisms such as acute phase protein activity, anorexia, lethargy, and fever are highly inflammatory, and thus may be ‘costly’ (Klasing & Leshchinsky, 1999). These responses are predicted (Cornet et al., 2016; Lee & Klasing, 2004), and have been shown, to be down-regulated in invasive populations of invertebrates (Cornet, Sorci & Moret, 2010; Wilson-Rich & Starks, 2010), sparrows (Lee, Martin & Wikelski, 2005), trout (Monzon-Arguello et al., 2014), and deer (Quéméré et al., 2015). Although constitutive innate defences (such as whole-blood phagocytosis of bacteria or yeast) also require substantial energy to activate (McDade, Georgiev & Kuzawa, 2016), glucose metabolism does not increase during phagocytosis in human neutrophils (Borregaard & Herlin, 1982). Thus, it is difficult to predict whether or not these defences are down-regulated in invaders, and further data are required.

The cane toad (Rhinella marina) was brought to Queensland, Australia from Hawai’i in 1935 (Turvey, 2009). Reflecting traits such as high fecundity and long-distance dispersal ability (Urban et al., 2008), toads have expanded their range into New South Wales (Easteal, 1981), the Northern Territory (Urban et al., 2008), and Western Australia (Rollins, Richardson & Shine, 2015). Populations have thus been exposed to local pathogens and parasites in Queensland for 81 years, whereas toads near the invasion front in Western Australia may be encountering novel pathogens for the first time. Surveys and common-garden experiments have shown that several phenotypic characteristics (including morphology, physiology, and behaviour) have diverged between populations from the ‘range core’ (QLD) and ‘invasion front’ (WA) (Brown et al., 2015; Gruber et al., 2017; Hudson, Brown & Shine, 2016). Due to differences in selection pressures for established populations (in the range core) vs expanding populations (at the invasion front), comparisons between them provide similar results to those expected between native and invasive populations.

Brown et al. (2015) compared the immunocompetence of captive-raised cane toads whose parents had been collected from QLD and WA. No significant difference was found in PHA-induced skin swelling, but WA toads exhibited higher bacteria-killing activity and phagocytosis than did QLD toads (Brown et al., 2015). This result suggests that bacteria-killing activity and phagocytosis may be favoured at the invasion front because these responses are less costly. However, Brown et al.’s study measured baseline levels of immune components, rather than the levels elicited by an in vivo immune challenge. One problem with comparing baseline levels of immune responses (such as phagocytosis and bacteria-killing activity) is the amount of variation across individuals in prior exposure to pathogens or antigens. Although individuals within the same population encounter the same types of infection, they may host pathogenic mutants of varying levels of virulence (McCahon et al., 1981). Measuring an immune response before and after experimental infection with agents like lipopolysaccharide (LPS, a bacterial endotoxin produced by Escherichia coli), and treating the change from baseline as the response variable, partially solves this problem by allowing comparisons within individuals through time, as well as among individuals. This is because baseline levels of an immune response (which are reflective of current pathogen load and previous exposure) are accounted for when analysing the response to the antigen. Experimental infection also provides the opportunity to study immune stimulation in a situation where the exact identity and dosage of the experimental antigen are known; this antigen has been isolated and purified, is no longer part of a live organism, and is not a nucleic acid (Gould, 2008), so mutation and replication are not possible. Thus, estimates of the strength of the immune response (duration, maximum response, and time to maximum response) are not confounded by the effects of differing pathogenic challenges. We applied the experimental infection method to clarify the relative phagocytic capabilities of toads from the range core vs toads from the invasion front.

Like the immune system, the hypothalamic–pituitary–adrenal (HPA) axis is stimulated by infection (Dunn & Vickers, 1994). The HPA axis is a major neuroendocrine system in vertebrates, and results in the release of glucocorticoids, which are closely associated with stress and activity (Janeway, Travers & Walport, 2001). Due to the suppressive effects of glucocorticoids on immune function (Alberts, Johnson & Lewis, 2002; Cooper, 2000; Fanning et al., 1998), glucocorticoids may regulate immune responses, preventing them from being elevated to a level that is harmful to the host (Ruzek et al., 1999; Stewart et al., 1988). In cane toads, associations between corticosterone and immune responses have previously been documented, with corticosterone having a negative effect on complement lysis activity, but a positive effect on leukocyte oxidative burst (Graham et al., 2012). Furthermore, toads with longer legs (a characteristic of WA toads) exhibited a reduced corticosterone response to stress (capture and confinement) than did their shorter-legged conspecifics (Graham et al., 2012). Because of this potential regulatory interaction and its relevance to immune function modulation in invaders, we also measured the effect of experimental infection on the glucocorticoid response.

Our study compared immune and glucocorticoid responses of wild-caught cane toads from both the invasion front (WA) and range core (QLD) populations after experimental injection with LPS. We expected infection with LPS to cause an increase in phagocytosis through stimulation of the immune system, regardless of population; we did not expect phosphate-buffered saline (PBS) to have an effect. Because infection affects the HPA axis (Dunn et al., 1989), we also expected LPS injection to increase corticosterone levels, regardless of population; we did not expect PBS to have an effect. At the population level, we expected differential effects of LPS on QLD toads and WA toads. If phagocytosis is indeed less costly than other immune responses, then we expected that individuals from the WA population may display higher levels of phagocytosis after injection with LPS than would individuals from the QLD population (as was seen by Brown et al. (2015) in common garden-bred toads). Because Graham et al. (2012) reported that cane toads of different leg lengths differed in glucocorticoid responses, we also expected a population difference in corticosterone levels (with the caveat that LPS injection may elicit a different response than that of capture and confinement, as used in the study by Graham et al. (2012).

Materials and Methods

Animal collection and husbandry

In May of 2016, specimens of R. marina were collected from two locations on opposite ends of the invasion transect. The eastern location (Cairns, QLD; 16.9186S 145.7781E) is the site of the initial release of toads into the wild in 1936 (Turvey, 2013). Toads did not arrive at the western location (Oombulgurri, WA; 15.1818S 127.8413E) until 2015; thus, this population represents the invasion front (Fig. 1). A total of 10 female toads per location were captured and transported to Middle Point, Northern Territory (12.5648S, 131.3253E), where they were maintained in a common setting for approximately one month before the experiment began. Only females were collected to eliminate possible sex effects, and for comparison with data on gene expression in female cane toads from a concurrent study. The experiment was conducted during the dry season in the Northern Territory, and thus toads were not breeding at this time. Toads from each location were divided into two groups of five: LPS-injection and PBS-injection (phosphate-buffered saline, control). Specimens were kept separate by their assigned group, and housed in large boxes in groups of two to three individuals. Mesh-covered openings in the boxes provided access to natural light, maintaining specimens on the Australian Central Time Zone light cycle and in outdoor temperatures (nocturnal temperatures ranged from 14 to 24.5 °C). Dust-free sawdust was used as a substrate, and plastic containers were provided for shelter. Water was changed daily, and crickets were distributed to each box every third day.

Figure 1 Map of cane toad distribution in Australia.

Current distribution of the cane toad throughout Australia. Toads were first introduced to Queensland (QLD) in 1935, and have since expanded their range into New South Wales, the Northern Territory (NT), and Western Australia (WA). Black diamonds indicate our toad collection sites: Cairns, QLD and Oombulgurri, WA.

LPS administration

Injections were performed using disposable 25-gauge needles with 1 mL syringes (Livshop, Rosebery, Australia). Each toad assigned to the LPS-injection group was size-matched (based on mass) with a toad from the same population that had been assigned to the PBS-injection group. Toads were injected with either 20 mg/kg body mass LPS (Sigma-Aldrich, Castle Hill, Australia) diluted in 100 μL PBS (Sigma-Aldrich, Castle Hill, Australia), or with an equal volume of pure PBS, to the dorsal lymph sac at 16:00 h. Average toad body mass was 131 g.

Blood sampling

Blood samples were taken for haemocytometry, white blood cell count differentials, and the phagocytosis assay. Cardiac punctures were performed using disposable 25-gauge needles with 1 mL heparinised syringes. Approximately 0.25 mL blood per individual was taken, and immediately transferred to a sterile 1.5 mL micro-centrifuge tube. Toads were not anesthetised; all samples were collected within 3 min of disturbance to the toad. This procedure was conducted on each toad twice: three days before and 14 days after injection, each time at 10:00 h. We allowed three days between blood collection and injection for toads to settle from the disturbance. A total of 14 days after injection were allowed for toads to mount an immune response; cellular and humoral immune responses have previously been shown to reach their maximum within this time frame in toads (Diener & Marchalonis, 1970).

Haemocytometry

To quantify the concentration of blood cells in each sample, 5 μL whole blood was diluted in 995 μL Natts–Herrick solution (Australian Biostain, Traralgon, Australia) and stored for 24 h at 4 °C. Then, blood cells were resuspended in the solution by inversion before 10 μL of the mixture was loaded into a haemocytometry chamber, and the numbers of erythrocytes (RBCs) and leukocytes were counted.

Counts of white blood cell differentials

Approximately 2 μL whole blood was used to prepare a smear that was then air-dried for an hour, and then stained with Diff-Quik (IHC World, LLC, Woodstock, MD, USA). After 24 h, cover slips were placed on each slide using a thin layer of mounting medium and samples were given another 24 h to dry. Slides were scanned at 100×, and the first 100 leukocytes seen were identified as basophils, eosinophils, neutrophils, lymphocytes, or monocytes. Percentages of each cell type (number of cells of each type divided by 100) were calculated. Because neutrophils are common phagocytes, the relationship between neutrophil percentage and phagocytosis was assessed.

Phagocytosis assay

We used a phagocytosis assay in which whole blood samples were stimulated by zymosan in the presence of luminol, generating luminescence (in relative light units, RLU) as a measure of phagocytosis (Martinez & Lynch, 2013). Whole blood was first diluted 1:20 in Amphibian Ringers solution, and 240 μL of the mixture was added to duplicate wells in a 96-well plate along with 30 μL luminol (Sigma-Aldrich, Castle Hill, Australia) and 10 μL zymosan (Sigma-Aldrich, Castle Hill, Australia). Another 240 μL of the sample was added to a control well along with 30 μL luminol and 10 μL PBS instead of zymosan. After the addition of zymosan, the 96-well plate was immediately inserted into a luminometer. Light emissions were recorded every 5 min for 200 min. The luminescence value in the control PBS well of each sample was subtracted from the luminescence values in the two corresponding zymosan wells to control for variations in light emissions between samples unrelated to the addition of zymosan; duplicates were then averaged together. Because there are multiple facets to the strength of an immune response (duration, maximum, and speed), phagocytosis was assessed via three response variables: mean luminescence across time points, maximum luminescence, and time to reach maximum luminescence. These three response variables were natural log-transformed for data normalization, then run through a principal component analysis to determine the best-fitting vector to represent all of the data in a single measure, called principal component 1 (PC1; Table 1). High PC1 values indicate high average luminescence, high maximum luminescence, and short time to maximum luminescence.

Table 1 Principal component analysis (PCA) loading values for three phagocytosis measures in cane toads.

Immune measure	PC1 (70.6%)	
Mean luminescence	0.97	
Max luminescence	0.93	
Time to max luminescence	−0.56	
Notes:

Loading values for principal component analysis (PCA) formed from three phagocytosis measures. Whole-blood phagocytosis in cane toads was assessed via mean luminescence across time points, maximum luminescence, and time to reach maximum luminescence. These response variables were natural log-transformed, and then run through a PCA.

Urine sampling

To obtain urine for corticosterone analysis, toads were lifted gently from their boxes and held over a plastic cup for up to 3 min. Urine was not collected from toads that did not urinate within this period. Urine was immediately transferred to a 2 mL snap-cap tube and stored at −20 °C. Urine sampling was conducted at seven time points during the experiment: three days (10:00 and 22:00 h) and two days (10:00 h) prior to injection, as well as six hours (22:00 h), one day (22:00 h), seven days (22:00 h), and 12 days (10:00 h) after injection. Samples were collected at two different times of day to incorporate periods of activity (22:00 h), when corticosterone levels are relatively high, and periods of inactivity (10:00 h), when corticosterone levels are lower (Jessop et al., 2014).

Creatinine assay

Creatinine concentration (μg/mL) was measured in every urine sample to standardise corticosterone levels by controlling for concentration of urine. Creatinine quantification was based on the Jaffe reaction in which creatinine turns orange in the presence of alkaline picrate. Briefly, 100 μL neat urine was mixed with 50 μL 0.75M sodium hydroxide (NaOH) and 50 μL 0.04N picric acid in duplicate on a 96-well plate. The plate was then incubated at room temperature for 15 min. Absorbance was measured with a plate reader (Biochrom Anthos 2010, Biochrom Ltd., Cambourne, UK) using a 405 nm measuring filter and a 620 nm reference filter.

Corticosterone assay

Urinary corticosterone metabolites were analysed using an enzyme-immunoassay that has been previously validated for cane toads (Narayan et al., 2012). Urinary corticosterone has been shown to lag behind plasma corticosterone by only 1 h (Narayan, Cockrem & Hero, 2013). The polyclonal corticosterone antibody (CJM06) and corresponding label (corticosterone conjugated with horseradish peroxidase [HRP]) were supplied by Smithsonian National Zoo (Washington, DC, USA). Briefly, high binding 96-well plates (Costar) were coated with 150 μl of coating buffer containing goat anti-rabbit IgG (GARG; 2 μg/mL). After 24 h, the coating solution was discarded and 200 μL of Trizma buffer solution rich in bovine serum albumin was added to each well and incubated for at least 4 h. Plates were washed five times and immediately loaded with 50 μL of standard, control, or neat urine sample, 50 μL of corticosterone-HRP (working dilution = 1:80,000), and 50 μL of corticosterone antibody (working dilution = 1:100,000). After incubating for 2 h at room temperature, plates were washed and 150 μL of substrate solution (1.6 mM hydrogen peroxide, 0.4 mM azino-bis(3-ethylbenzthiazoline-6-sulphonic acid) in 0.05 M citrate buffer, pH 4.0) was added to each well. The plate was incubated at room temperature for 45 min, and absorbance was quantified using a 450 nm measuring filter and a 620 nm reference filter. All samples were analysed in duplicate and hormone concentration is expressed as ng/μg creatinine. Urine corticosterone metabolite concentrations were natural log-transformed to meet model assumptions.

Statistical analyses

All statistical analyses were performed using JMP Pro 11.0 (SAS Institute, Cary, NC, USA). We first subtracted each individual’s pre-injection PC1 score from its post-injection PC1 score, and then analysed the differences using a linear mixed model containing red blood cell concentration, population (QLD vs WA), and treatment (LPS injection vs PBS injection) as fixed effects. The interactive effects of population and treatment were also included in the model. We used this model to assess variation in PC1 scores, as well as variation in neutrophil percentages. Changes in PC1 were also tested for correlation with changes in neutrophil percentage due to the putative role of neutrophils in phagocytosis. A p-value less than 0.05 was used as our criterion for statistical significance of an effect or relationship.

Hormone data were analysed using a linear mixed model containing population, treatment, and days post-injection (−3.25, −2.75, −2.25, 0.25, 1.25, 7.25, and 12.25 DPI) as fixed effects. The interactive effects of population, treatment, and DPI were also included in the model. Because several repeated measures were taken for each toad, we also included individual ID as a random effect in the model. A p-value less than 0.05 was required to call significance of an effect or relationship.

Ethics

Research was conducted in accordance with rules set for by the University of Sydney Animal Ethics Committee. Ethics application was approved under the project number 2016/1003.

Results

Phagocytosis

We found an effect of LPS challenge treatment on phagocytosis PC1 (treatment effect p = 0.02). LPS-injected toads exhibited a greater increase in phagocytosis than did their PBS-injected counterparts after injection (Fig. 2). RBC count also had a significant effect; toads whose blood samples were more concentrated with RBCs exhibited higher levels of phagocytosis. However, population had no significant effect, indicating similarity in the two populations’ phagocytic response to LPS challenge (Table 2).

Figure 2 Phagocytic responses of cane toads to lipopolysaccharide (LPS).

Phagocytosis curves of cane toads (Rhinella marina) from (A) Queensland (QLD) and (B) Western Australia (WA) both before and after injection with either lipopolysaccharide (LPS) or phosphate-buffered saline (PBS). The average of all samples within a treatment group at each time point (N = 5) was calculated to produce the points on the graph. (C) Difference in mean luminescence values between pre-injection and post-injection readings of each population and treatment group.

Table 2 Cane toad phagocytosis linear mixed model.

Source	Estimate	Standard error	95% CI	DF	F ratio	p	
RBC	0.01	0.004	[0.004, 0.02]	1,15	8.70	0.01	
Population	0.63	0.33	[−0.07, 1.34]	1,15	3.61	0.08	
Treatment	−0.87	0.34	[−1.58, −0.15]	1,15	6.64	0.02	
Population × treatment	−0.06	0.33	[−0.77, 0.65]	1,15	0.03	0.85	
Notes:

Effect of source population and treatment (LPS or PBS) on phagocytic activity in blood of the cane toad, Rhinella marina. Each individual’s pre-injection PC1 score was subtracted from its post-injection PC1 score, and differences were analysed using a linear mixed model containing red blood cell concentration (RBC), population (Queensland vs Western Australia), and treatment (LPS injection vs PBS injection) as fixed effects. Significant effects are in bold.

White blood cell differentials

Similar to PC1, a strong treatment effect was seen on change in neutrophil percentage (p = 0.0052). Across both populations, LPS-injected toads exhibited a greater increase in the percentages of neutrophils in their blood than did their PBS-injected counterparts after injection (Fig. 3A). Across treatments, the change in neutrophil percentage before vs after injection was positively correlated with the change in PC1 before vs after injection (p = 0.037, R2 = 0.23; Fig. 3B).

Figure 3 Neutrophil responses of cane toads to lipopolysaccharide (LPS).

(A) Changes in the percentages of neutrophils (as induced by injection of either lipopolysaccharide [LPS] or phosphate-buffered saline [PBS]) across two treatment groups of cane toads (Rhinella marina) from two populations (invasion front site in Western Australia [WA] and range core site in Queensland [QLD]). The percentage of neutrophils in the toad’s blood pre-injection was subtracted from the percentage of neutrophils in the same toad’s blood post-injection. The average of the difference between pre-injection and post-injection of all samples within a treatment group at each time point (N = 5) was calculated to produce the points on the graph. Error bars indicate standard error. (B) Positive correlation between the changes in neutrophil percentage and PC1 before vs after injection.

Urinary corticosterone metabolites

A strong treatment × days post-injection (DPI) effect was observed on urinary corticosterone levels (Table 3). Corticosterone increased over time after injection in PBS-injected toads, but decreased over time after injection in LPS-injected toads (Fig. 4). However, we found no significant difference between populations.

Table 3 Cane toad urinary corticosterone linear mixed model.

Source	Estimate	Standard error	DF	F ratio	p	
Population	−0.14	0.17	1,14	0.67	0.4253	
Treatment	−0.04	0.17	1,14	0.05	0.8311	
DPI	−0.02	0.02	1,41	1.83	0.1839	
Population × treatment	0.02	0.17	1,14	0.02	0.8909	
Population × DPI	−0.02	0.02	1,41	1.12	0.2964	
Treatment × DPI	−0.05	0.02	1,41	10.53	0.0023	
Population × treatment × DPI	0.02	0.02	1,41	0.89	0.3516	
Notes:

Effects of each explanatory variable on corticosterone levels of cane toads after injection with either lipopolysaccharide (LPS) or control (phosphate-buffered saline, PBS). Natural log-transformed corticosterone values were analysed in a linear mixed model with population (Queensland vs Western Australia), treatment (LPS injection vs PBS injection), and days post-injection (DPI) as fixed effects; their interactive effects were also assessed. Significant effects are in bold.

Figure 4 Urinary corticosterone responses of cane toads to lipopolysaccharide (LPS).

Changes in urinary corticosterone levels across two treatment groups of cane toads (Rhinella marina) from two populations (invasion front site in Western Australia [WA] and range core site in Queensland [QLD]). Toads were injected with either lipopolysaccharide (LPS) or phosphate-buffered saline (PBS). Error bars indicate standard error, and lines are fitted by linear regression to data from each treatment group.

Discussion

Injection with LPS evoked up-regulation of immune responses in cane toads from both expanding (invasion front) and established (range core) populations. Two weeks after injection with LPS, toads had increased circulating levels of neutrophils, as well as elevated phagocytic ability. Control toads injected with PBS showed no significant changes in these immune measures. A positive correlation between phagocytosis levels and neutrophils was expected, as neutrophils are the most abundant phagocytic cells in the blood (Summers et al., 2010; Wilgus, Roy & McDaniel, 2013; Wright, 2001). Indeed, toads that increased neutrophil production the most also increased phagocytic ability the most, indicating an increased stimulation of neutrophil-mediated phagocytosis by LPS. Thus, LPS is an effective stimulant of the phagocytic response in cane toads via their neutrophils.

Blood samples that were more concentrated with RBCs exhibited higher levels of phagocytosis. However, RBCs had no effect on neutrophil abundance (p = 0.55). Thus, the effect could have potentially arisen through luminescent properties of the RBCs themselves (autofluorescence; Emmelkamp et al., 2003).

Contrary to our predictions, toads from the invasion front did not exhibit stronger phagocytic responses to LPS exposure than did toads from the range core; there was no significant difference between populations. Our predictions were based upon the possibility that neutrophil-mediated phagocytosis poses a lower energetic cost than other immune responses, and hence may be favoured in toads at the invasion front undergoing enemy release. Brown et al. (2015) reported higher baseline levels of neutrophil-mediated phagocytosis in common garden-raised toads from WA, consistent with the hypothesis that neutrophil-mediated phagocytosis provides an inexpensive way for WA toads to retain some immunocompetence without expending much energy. Thus, we expected phagocytosis to be favoured in the wild-collected adult toads from WA as well, even with the change in methodology (injection with LPS vs measurement of baseline levels). However, LPS is a common bacterial antigen, and the lack of difference may reflect an equal importance of combating severe bacterial infections across all populations. There may be more than a billion species of bacteria worldwide, and they are found across all environments (Dykhuizen, 1998). Because bacteria are able to tolerate many types of abiotic extremes (Dykhuizen, 1998), it is likely that the cane toad’s entire Australian range is home to rich bacterial communities. Although microbial species richness follows an aridity gradient (Yabas, Elliott & Hoyne, 2015) and the particular species of bacteria present at opposite ends of the range may differ, standing innate immune defences such as neutrophils are unspecialised (Janeway, Travers & Walport, 2001), and thus may not differ across populations based on changes in the bacterial species encountered.

However, it is also possible that toads from invasion front vs range core populations differ in phagocytic activity (as reported by Brown et al. (2015)), but that those differences were not apparent in our study. Our sample sizes were limited to N = 5 per treatment per population. When a null hypothesis is not rejected (such as in our study, in which a population-level difference was not found), confidence intervals for the effect size are recommended in place of retrospective power analysis to check for validity (Steidl, Hayes & Schauber, 1997). Our 95% CI for population as a predictor include zero within their range (Table 2); this indicates that effect could be zero, and thus rejection of the null hypothesis is justified. The same is seen in the 95% CI for the interactive effect of population and treatment. Sample sizes were low and it is possible that this may not have been sufficient to uncover a population-level difference; however, there are also several biological explanations for the lack of population difference.

Maintaining toads from both populations in a common captive setting, and feeding them the same food, may have eliminated differences in their gut microbiota (Riddell et al., 2014), which in turn influences immune function (Carpenter et al., 2014). Additionally, toads from WA are constantly dispersing into novel environments (Urban et al., 2008) where they will encounter unfamiliar pathogens and parasites. Our study utilised wild-caught toads; prior to our collection, those from WA may have expended more of the energy allocated to neutrophil production and activity than did those from QLD (Brown et al., 2015), potentially resulting in a diminution of the intrinsically stronger phagocytic response. This idea is supported by a previous study conducted on wild-caught cane toads from the Northern Territory, where the toads were radio-tracked to quantify their movement distances before their immune responses were surveyed (Brown & Shine, 2014). Toads that travelled longer distances exhibited decreased standing innate defences, such as neutrophils, compared to less mobile toads (Brown & Shine, 2014). Although these toads were not collected from the same areas as those in our experiment, ‘more mobile’ may be a reasonable proxy for WA, and ‘less mobile’ may represent QLD. If this is the case, a lifetime use of neutrophils and other standing innate effectors may explain why WA toads do not retain the stronger neutrophil responses with which they are born.

Prior energy expenditure is not the only variable that could have obscured a potential difference in phagocytosis levels between populations in our study. Our data suggest that the toads’ immune responses may have also been dampened by abiotic conditions. On average, the PBS-injection groups from both populations exhibited a decrease in levels of phagocytosis after injection. This decrease may be due to temperature; our experiments were conducted in mid-July to early August, when nocturnal temperatures fall as low as 14 °C. Low temperatures have immunosuppressive effects on amphibians and other ectotherms (Maniero & Carey, 1997; Pxytycz & Jozkowicz, 1994; Raffel et al., 2006). Neutrophils and phagocytic activity decrease initially during winter, but return to baseline levels as amphibians acclimate to seasonal temperatures (Raffel et al., 2006). Such seasonal effects in our toads may have masked population differences.

Stress may have also influenced our results. Corticosterone, the primary stress glucocorticoid hormone in amphibians, can induce differential suppression and activation of immune components (Stier et al., 2009). Cane toads exhibit an acute stress response to capture in which their corticosterone levels initially increase (Graham et al., 2012), but decline back to baseline after seven days of confinement (Narayan, Cockrem & Hero, 2011); our toads were held in captivity for one month prior to our study. Although our study showed that PBS-injected toads increased in corticosterone levels after injection, we observed the opposite effect in LPS-injected toads. This finding was unexpected; in mammals, glucocorticoids increase during infection, possibly to regulate the immune response and prevent excessive inflammation (Hawes et al., 1992; Ruzek et al., 1999; Stewart et al., 1988; Webster & Sternberg, 2004). However, one month in captivity, frequent handling, injection with antigens, and cardiac puncture may have induced a state of chronic stress in the toads (Wingfield & Romero, 2001). Adrenal activity is less predictable during chronic stress, and thus the direction of the corticosterone response differs across taxa and stressor types (Dickens & Romero, 2013). Because the amount and nature of handling was the same for toads across all treatments, their stress states were likely similar; however, infection apparently made a difference for the LPS-injected toads. Some chronically stressed animals exhibit a decrease in corticosterone levels after infection (Cyr et al., 2007), thereby avoiding the suppressive effects of chronically high corticosterone on immune function. Amphibian immune defences are particularly sensitive to glucocorticoids, and increases in these hormones can raise amphibians’ susceptibility to disease (Rollins-Smith, 2001). Additionally, corticosterone is not the only mechanism by which stress can suppress immunity. Stress may also cause leakage of gut microbiota into the bloodstream, triggering immune responses; this results in a lower number of unoccupied effectors (Lambert, 2014; Saunders et al., 1994).

Studies on phagocytosis in cane toads have ignored differences in the speed, rather than in the strength, of this immune response. Although phagocytosis levels did not differ significantly between populations, the speed at which the observed up-regulation occurred might have differed. A previous study found that LPS injection triggered a larger metabolic increase after 24 h in QLD toads than in WA toads (Llewellyn et al., 2012). Conceivably, part of this increased metabolic activity seen among QLD toads could have involved increased production of immune components. Our post-injection immune assays were conducted two weeks after LPS injection, by which time toads from both populations had up-regulated immune responses to the same extent. Future studies could explore this question by measuring phagocytosis 24 h after LPS exposure and monitoring changes in phagocytosis across a shorter time frame.

Conclusion

In this study, we tested phagocytosis in cane toads using the same cell quantification methods and activity assay as those of Brown et al. (2015), but we took repeated measurements before and after injecting LPS in vivo. Our study confirmed that LPS stimulates phagocytosis; however, we did not detect a population-level difference in phagocytosis levels (as had been found in the previous study). Each experiment introduces its own unique confounds; the previous study did not account for inter-individual variation, and ours could not account for differences in environmental effects prior to collection. To definitively compare levels of phagocytosis between individuals from invasion front vs range core populations, a more robust experimental design would employ the experimental antigen methodology simultaneously on wild-caught toads, and on captive-bred toads raised in a common setting from each population.

Supplemental Information

Supplemental Information 1 Raw data.

Click here for additional data file.

Additional Information and Declarations

Competing Interests

Author Contributions

Animal Ethics

Data Availability

Lee A. Rollins is an Academic Editor for PeerJ.

Daniel Selechnik conceived and designed the experiments, performed the experiments, analysed the data, contributed reagents/materials/analysis tools, wrote the paper, prepared figures and/or tables.

Andrea J. West conceived and designed the experiments, performed the experiments, reviewed drafts of the paper.

Gregory P. Brown conceived and designed the experiments, performed the experiments, analysed the data, reviewed drafts of the paper.

Kerry V. Fanson conceived and designed the experiments, reviewed drafts of the paper.

BriAnne Addison conceived and designed the experiments, reviewed drafts of the paper.

Lee A. Rollins conceived and designed the experiments, contributed reagents/materials/analysis tools, reviewed drafts of the paper.

Richard Shine conceived and designed the experiments, contributed reagents/materials/analysis tools, reviewed drafts of the paper.

The following information was supplied relating to ethical approvals (i.e. approving body and any reference numbers):

Research was conducted in accordance with rules set for by the University of Sydney Animal Ethics Committee (Approval number: 2016/1003).

The following information was supplied regarding data availability:

The raw data has been provided as Supplemental Dataset Files.

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
