# Peer review of "Effects of invasion history on physiological responses to immune system activation in invasive Australian cane toads"

_PeerJ, doi:10.7717/peerj.3856_

## Round 0.1 · original submission · Minor Revisions

· Academic Editor

Minor Revisions

Both reviewers suggest that this manuscript should be acceptable with only Minor Revisions; a view with which I concur. Please make your revisions in line with the reviewer comments.

Thank you for your submission.

Reviewer 1 ·

Basic reporting

Overall, I thought the article was well written and set up very well with previous research. I only have a couple of comments:

Ln 70: "These responses are predicted (and have been
70 shown) to be down-regulated in invasive populations (Lee & Klasing 2004; Lee et al. 2005)." Some examples of types of immune responses and what taxa would be helpful here.

Ln 71: What does standing immune responses mean? Does this mean constitutive?

Experimental design

Ln 194: I assume the authors mean 30 ul of Luminol.

Validity of the findings

Ln 115: Add the direction of the relationship.

In the table, the effect of RBCs is significant. Is this due to the autoflorensce of the RBCs? Just a brief sentence explaining this may be helpful for those not familiar with the technique.

Low sample size, but it is acknowledged by authors and conclusions are framed appropriately.

·

Basic reporting

The manuscript is clear and very well written. Title and abstract are both informative. Introduction supports the study and adequately refers to pertinent literature. However, the logic flow of the introduction should be improved to explain more effectively the main hypothesis of the study (see specific comments below). The structure of the manuscript conforms to PeerJ standards. Figures and tables are all nice and properly labelled. Raw data are provided.

Experimental design

The research is within the scope of the journal and the research questions are quite clearly stated (but see specific comments). The experimental protocol has been carefully selected not only to address the specific questions of the study but also to propose a novel effective procedure to investigate immune pathways. Since a previous work (Brown et al. 2015) which investigated the same aspect (immune response) in the same two populations of cane toad did not account for inter-individual variation, the study also represents a further step in our comprehension of invaders’ eco-immunology. The research protocol is rigorous and numerous details are provided to replicate the study.

Validity of the findings

Although the main prediction of the study (that is higher less costly immune response in toads from the invasion front) is not confirmed by the results, the findings of the manuscript are still valid with several suggested explanations to support the null outcome. Also, limitations of the study are clearly recognized (e.g. it did not “account for differences in environmental effects prior to collection”, L393-394) and future more advanced studies are proposed. Data is controlled and statistics used is adequate. The sample size is small and this could partially obscure the results but the limitation is well recognized in the manuscript. Conclusions are limited to supporting results and speculations are recognized as such. However, I feel that the second part of the discussion is a little bit too long (one or two paragraphs should be shortened or removed) and speculative. Only most valid speculations should be incorporated (see specific comments).

Additional comments

General comments

In this study, immune response and glucocorticoid response are experimentally induced in two Australian invasive populations of cane toads (one from the core and one from the periphery of the invasion) by injecting an antigen lipopolysaccharide. The study is very interesting and well supported by the exceptional literature regarding this invasive species. Although the results are not fully conclusive (no difference detected between the populations), the robustness of the protocol and the novelty of an immune response measurement in vivo make its publication recommended. Although I do not have any major revision to recommend, I have added some comments below that I hope could improve the readability of the manuscript, especially in the introduction section.

Specific comments

L38 - add “wild-caught” before “toads”?

L55-75 - The first part of the introduction, although logic and well structured, is a little bit misleading. The authors state correctly that a down-regulation of the costly systemic inflammatory response is expected in invasive populations (or in the populations at the invasion front); however eco-immunology theory predicts also that less costly immune pathways should be up-regulated by invasive populations or species (see for example “humoural” response in Lee and Klasing 2004). The authors, later in the text, hypothesize exactly this up-regulation (also in accordance to a similar study conducted by Brown et al. 2015 that compared the same two invasive populations). Therefore, they should clarify here (and not later, at L89-91) that two divergent eco-evolutionary scenarios (downregulation of costly immune pathways and upregulation of less costly immune pathways) are expected in invasive (or invasion front) populations. This should be helpful, especially for a general reader, to comprehend the aim of the study, its assumptions and also the main prediction reported at L129-132.

L73 – “in human neutrophils” Could you find more references, not necessarily in humans, that support this?

L76-77 “To Australia from Hawaii” - The cane toad did not arrive in Australia directly from the native range (South America) but rather it was moved from one area to another (e.g. Hawaii) until it established in Australia. It means that during its invasion history, it has been exposed to many different groups and sources of pathogens. Could have this promoted a strong less costly response of the immune system that become canalized (less plastic) in the invasive Australian populations? Since you are not comparing a native pop with an invasive population but two invasive populations, this should be somewhere stated in the manuscript, maybe in the discussion.

L90-91 I would suggest “less costly” instead of “not costly”. Any immune response should have a cost, although minimal.

L 104 “because this antigen is no longer part of a live organism, mutation and replication are not possible”. Any reference for this?

L107-108 “We applied this experimental method to clarify immune function in different populations of invasive cane toads.” This sentence is too generic, please rephrase.

L139-140 Could you please add the geographic coordinates for the two sites of capture (Cairns and Oombulgurri)? Also, a map of Australia with the cane toad invasion, although not essential, could help the reader to visualize introduction point and current invasion front.

L290-291 “Contrary to our predictions, responses to LPS exposure did not differ significantly between toads from the invasion front population versus the range core population”. Your prediction was more specific than that because you were expecting an up-regulation; please rephrase.

L291-293 “Our predictions were based upon the energetic costs associated with mounting immune responses while undergoing enemy release.” This sentence is too generic, please rephrase. Also, as many different types of immune responses can occur and some are not costly, I would suggest being more specific across the whole manuscript for example distinguishing between systemic inflammatory responses and humoral responses (as stated in Lee and Klasing 2004).

L310-319 This could be summarized in only two sentences.

L 330 “More mobile toads”. Please clarify this for a general reader. Do they move more frequently? Do they move longer?

L 370-376 Since Brown et al. 2015 did not detect any sex effect, this paragraph could be removed because it does not add anything essential to the interpretation of your results.

L574 Please fix reference (space issue)

In the tables. Significant results in bold? Also, p-value is not defined anywhere in the manuscript.

---

## Round 0.2 · accepted · Accept

· Academic Editor

Accept

Thanks for your clear changes to the manuscript which I now find acceptable for publication in PeerJ.